# Anomalous Kondo resonance mediated by semiconducting graphene nanoribbons in a molecular heterostructure

Yang Li[1,2], Anh T. Ngo[3], Andrew DiLullo[1], Kyaw Zin Latt[1,2], Heath Kersell[1,2], Brandon Fisher[1], Peter Zapol [3], Sergio E. Ulloa [2] & Saw-Wai Hla[1,2]

Kondo resonances in heterostructures formed by magnetic molecules on a metal require free host electrons to interact with the molecular spin and create delicate many-body states. Unlike graphene, semiconducting graphene nanoribbons do not have free electrons due to their large bandgaps, and thus they should electronically decouple molecules from the metal substrate. Here, we observe unusually well-defined Kondo resonances in magnetic molecules separated from a gold surface by graphene nanoribbons in vertically stacked heterostructures. Surprisingly, the strengths of Kondo resonances for the molecules on graphene nanoribbons appear nearly identical to those directly adsorbed on the top, bridge and threefold hollow sites of Au(111). This unexpectedly strong spin-coupling effect is further confirmed by density functional calculations that reveal no spin–electron interactions at this molecule-gold substrate separation if the graphene nanoribbons are absent. Our findings suggest graphene nanoribbons mediate effective spin coupling, opening a way for potential applications in spintronics.

[1] Center for Nanoscale Materials, Nanoscience and Technology Division, Argonne National Laboratory, Lemont, IL 60439, USA. [2] Department of Physics and Astronomy, Nanoscale and Quantum Phenomena Institute, Ohio University, Athens, OH 45701, USA. [3] Materials Science Division, Argonne National Laboratory, Lemont, IL 60439, USA. Correspondence and requests for materials should be addressed to S.-W.H. (email: hla@ohio.edu)

Atomically precise semiconducting graphene nanoribbons are one-dimensional graphene strips[1–3] with varying bandgaps depending on their width and length[4–8]. Unlike graphene, which has a semimetallic character, graphene nanoribbons are more suited for applications ranging from electronic and optoelectronic devices to sensors due to their semiconducting gap. Among different types of graphene nanoribbons, atomically precise armchair edge graphene nanoribbons (AGNRs) can be synthesized on surfaces using basic chemical ingredients[2, 3, 9, 10], which opens potential bottom-up fabrication of graphene nanoribbons-based devices. For example, lateral heterojunctions can be formed by fusing grapheme nanoribbons with different widths thereby realizing band-gap engineering at the atomic limit[11]. For potential applications of AGNRs, it is vital to explore vertically stacked heterostructures formed by molecules of interest on top of AGNRs. So far, single molecule level studies of individual molecules adsorbed on AGNRs have yet to be reported.

Porphyrin group molecules with caged metal atoms[12–18] are known to exhibit Kondo effect[19–21], which is generated by many body correlations between the magnetic moment of the molecule and the spins from the free electrons in the substrate[22]. A porphyrin-based magnetic molecule, TBrPP-Co, has a cobalt atom caged at its centre, and has a net spin ½ with the spin density localized at the $3d^7$ Co(II) state[18]. TBrPP-Co exhibits Kondo effect when it is adsorbed on metal surfaces such as Cu (111)[18]. If the magnetic moment of the molecule is electronically decoupled from the free electron host of the substrate, then the Kondo effect is not expected to occur. Semiconducting AGNRs grown on Au(111) surface may be useful as buffers to separate the magnetic molecular orbitals from the substrate because of their sizable band gaps. Moreover, it is known that there is no charge transfer between the AGNR and Au(111)[23] to induce additional charges in AGNR so that an effective electronic decoupling should be expected.

In the following, we investigate electronic and spintronic properties of vertically stacked heterostructures formed by TBrPP-Co molecules adsorbed on AGNRs on a Au(111) surface using low-temperature scanning tunnelling microscopy and spectroscopy at the atomic limit supported by density functional theory (DFT) calculations. The geometrically relaxed DFT calculations reveal a large vertical distance (7.5 Å) between the molecule and the underlying Au(111) surface when an AGNR is sandwiched in between. This separation would effectively decouple the TBrPP-Co from the Au(111) surface electronically. Yet, we have detected strong Kondo resonances on TBrPP-Co adsorbed on AGNR with three different Kondo temperatures tracing atomic details of the adsorption sites on the underlying Au(111) surface. This suggests that AGNRs are rather effective in mediating the interaction of the molecule magnetic moment with electronic spins in gold. Such mediation of spin interactions is reminiscent of the transparency of graphene seen on Cu(111) surfaces[24], which projects a long-range electronic density with copper character away from the surface, and is uniquely made evident here by the subtle Kondo probe.

## Results

**AGNR/Au(111).** We use DBBA (10,10′-dibromo-9,9′-bianthryl) precursor molecules[2, 3, 25, 26] as basic ingredients to synthesize AGNRs on an atomically clean Au(111) surface in an ultrahigh vacuum environment. After deposition of the DBBA[26] onto Au (111) surface, heating the sample to ~200 °C breaks up the bromines of DBBA and links the molecules to form long polymer chains. Further heating to a higher temperature of ~400 °C leads to the formation of AGNRs (Fig. 1a, b). The long axes of the AGNRs mostly align along the [110] surface close-packed row directions of Au(111) although they often orient to other surface directions. The smallest width of AGNR has seven linked carbon atoms (7-AGNR). By increasing the DBBA coverage on Au(111), wider AGNRs such as 14, 21 and 28 AGNRs can be synthesized[8]

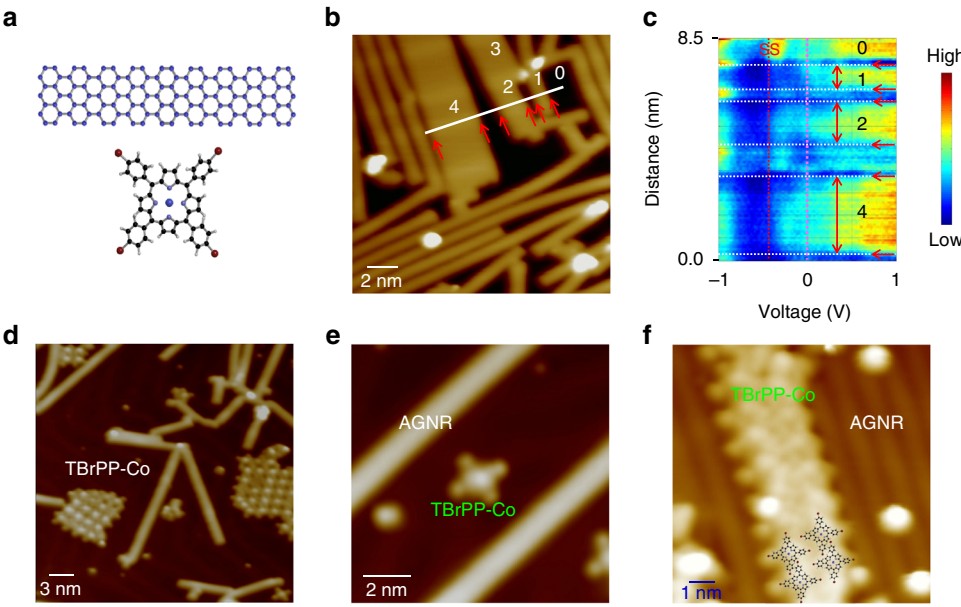

**Fig. 1** TBrPP-Co/AGNR heterostructures. **a** Structure of 7-AGNR (*top*) and TBrPP-Co (*bottom*). **b** A scanning tunneling microscope (STM) image showing AGNRs with various widths on Au(111) (16.5 × 16.5 nm², $I_t = 1 \times 10^{-10}$ A, $V_t = 1$ V). Here, 1, 2, 3 and 4 label 7, 14, 21 and 28-AGNR, respectively. **c** d$I$/d$V$ spectroscopy scan as a function of distance measured along a white line in **b**. Regions 1, 2 and 4 label corresponding AGNRs in **b** and 0 is Au(111). The *arrows* indicate the edge of AGNRs, while the *red dot line* marks the Shockley surface state (SS) on-set of Au(111). (Tip set-point: $I_0 = 1.0 \times 10^{-10}$ A, $V_0 = 1.0$ V). **d** An STM image shows TBrPP-Co clusters grow in between AGNRs on Au(111) (31 × 31 nm², $I_t = 3 \times 10^{-11}$ A, $V_t = -0.1$ V). **e** STM image of a single TBrPP-Co adsorbs on Au(111) between two AGNRs (11 × 11 nm², $I_t = 5 \times 10^{-11}$ A, $V_t = -0.2$ V). **f** STM image of TBrPP-Co molecular chains formed on top of AGNR (10 × 10 nm², $I_t = 1 \times 10^{-11}$ A, $V_t = 0.4$ V)

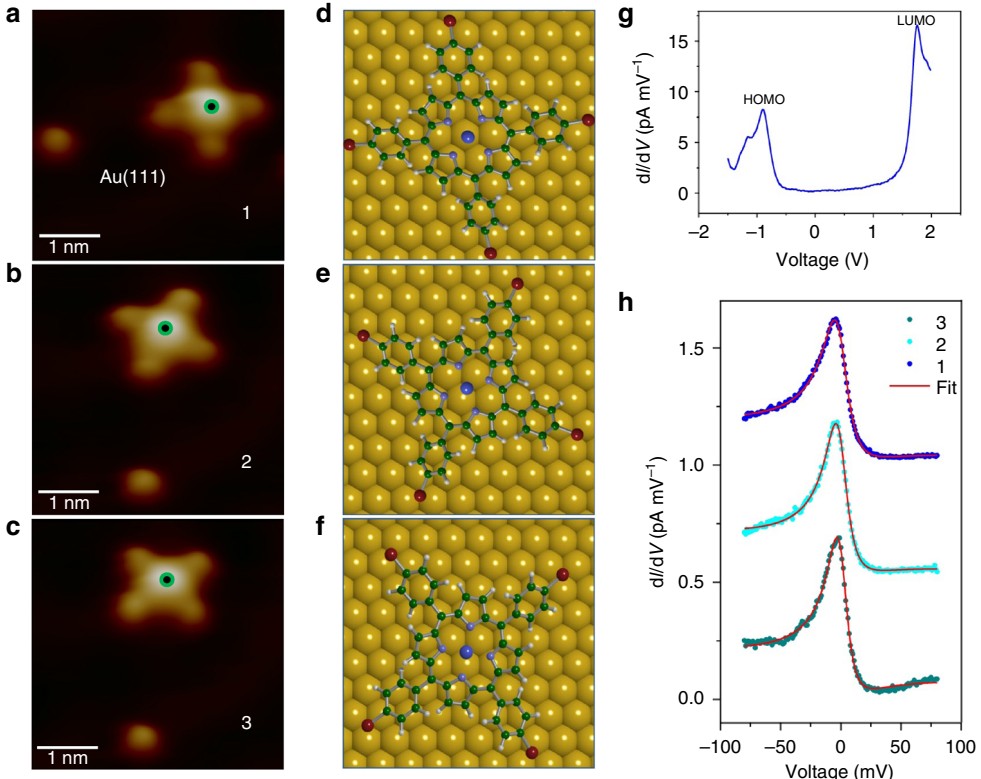

**Fig. 2** Properties of TBrPP-Co/Au(111). **a**–**c** Three orientations of TBrPP-Co on Au(111) induced by STM manipulation labelled as 1, 2 and 3, respectively. The protrusion next to the molecule is used as a landmark (4.2 × 4.2 nm$^2$, $I_t = 1 \times 10^{-10}$ A, $V_t = -1.0$ V). *Green* dots indicate where the Kondo effect is measured. **d**–**f** DFT adsorption structures of TBrPP-Co corresponding to 1, 2 and 3, (**a**–**c**), respectively. **g** A d$I$/d$V$–$V$ spectroscopic data of TBrPP-Co on Au(111) showing HOMO–LUMO orbitals (tip set-point: $I_0 = 5 \times 10^{-11}$ A, $V_0 = -1.0$ V). **h** High-resolution d$I$/d$V$ spectra of TBrPP-Co/Au(111) corresponding to three adsorption geometries, 1, 2 and 3, revealing a Kondo resonance ($q = 0.37$, 0.36 and 0.40 for 1, 2 and 3, respectively. Tip set-point: $I_0 = 1.0 \times 10^{-10}$ A, $V_0 = -0.1$ V. Spectra are vertically offset by 0.5 pA/mV for clarity)

(Fig. 1b and Supplementary Fig. 1). Electronic structures of AGNRs on Au(111) are measured by means of d$I$/d$V$ tunnelling spectra as functions of distance and bias (Fig. 1c). The AGNRs are known to screen/depopulate the Shockley surface state (SS) of Au(111)[23]. Underneath the AGNRs, the SS onset energy is shifted towards the surface Fermi level (i.e., 0 V), and this energy shift is found to be dependent on the width of the AGNRs: the larger the width, the more the SS shifts towards the Fermi level (Supplementary Fig. 2 and Supplementary Note 1). Moreover, the distance-dependent d$I$/d$V$ spectra reveal the edges of the AGNRs with a larger gap appearing as streaks in Fig. 1c. This effect is caused by the strong bonding of the hydrogen termination at the AGNR edges (Supplementary Fig. 3 and Supplementary Note 2). As H binds strongly to the AGNR edges, it removes electronic density away, which effectively opens the gap locally. AGNRs are known to weakly adsorb on Au(111) and they have recently been shown to exhibit superlubricity[27]. Tunnelling spectroscopy and angle resolved photoemission spectra measurements[23] also highlight the absence of interfacial charge transfer for AGNRs on Au(111). In agreement with these findings, our DFT calculation shows no significant interfacial charge transfer, and AGNR behaves as an insulating layer on Au(111) due to the bandgap.

**TBrPP-Co/Au(111)**. The molecule used for this investigation, TBrPP-Co, has a cobalt (Co) atom caged at the centre of the porphyrin unit[28, 29] (Fig. 1a). When TBrPP-Co is deposited onto Au(111) sparsely populated with AGNRs, the molecules adsorb only on the bare Au(111) surface areas (Fig. 1d, e) indicating that the molecule–surface interaction is stronger than the molecule–AGNR interaction (Supplementary Fig. 4). The TBrPP-

Co/AGNR/Au(111) vertically stacked heterostructures can be successfully formed when the molecules are deposited onto densely packed AGNRs on Au(111) (Fig. 1f). The size of the TBrPP-Co is larger than the width of 7-AGNR, and most of the molecular clusters are formed on larger width AGNRs such as 21 and 28 AGNRs.

To highlight the influence of AGNR on the molecules, we first investigate the properties of TBrPP-Co on the Au(111) surface. TBrPP-Co adsorbs with a planar conformation on Au(111), and appears in three rotation geometries (Fig. 2a–c) and (Supplementary Figs. 5 and 6, and Supplementary Note 3). The DFT + $U$ calculations (see Methods section) reveal each rotation as due to different adsorption sites where the Co centre of the molecule is positioned on top, bridge or threefold hollow sites of Au(111) (Fig. 2d–f), respectively. Here, the molecule is located 3.46 Å above the Au surface and it is physisorbed. The electronic structure of TBrPP-Co is determined by means of large bias range d$I$/d$V$–$V$ tunnelling spectroscopy, which yield the highest occupied and lowest unoccupied molecular orbitals (HOMO and LUMO) for the molecule adsorbed at a surface hollow site as −0.9 and +1.8 V, respectively (Fig. 2g), providing a HOMO–LUMO gap of 2.7 V. For the TBrPP-Co adsorbed on bridge and top surface sites, the HOMO–LUMO gaps are found as 2.6 and 2.54 V, respectively (Supplementary Fig. 7 and Supplementary Note 3).

The spin-polarized (SP) DFT calculations for the TBrPP-Co adsorbed on Au(111) show no interfacial charge transfer between the molecule and Au(111), and its spin density remains at the 3d$^7$ Co(II) centre. As expected, Kondo resonances are observed as Fano shape peaks around the Fermi level in high-resolution d$I$/

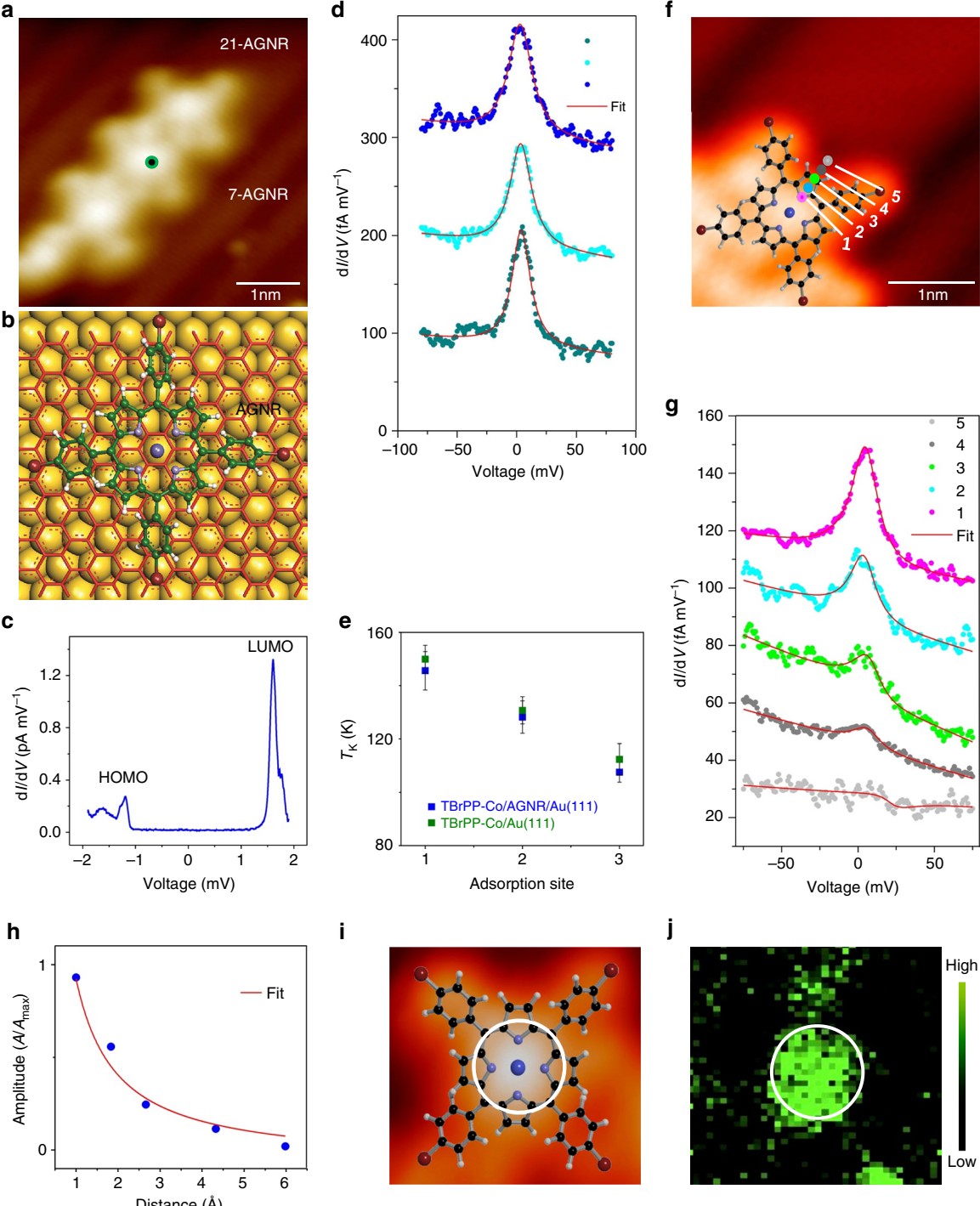

**Fig. 3** Properties of TBrPP-Co/AGNR/Au(111). **a** STM image of TBrPP-Co on AGNR ($4.5 \times 4.5$ nm$^2$, $I_t = 3 \times 10^{-12}$ A, $V_t = 1$ V). **b** Adsorption site of TBrPP-Co on AGNR. **c** d$I$/d$V$–$V$ spectroscopy of TBrPP-Co on AGNR/Au(111) shows a flat bottom at the HOMO–LUMO gap (tip set-point: $I_0 = 1.0 \times 10^{-11}$ A, $V_0 = 1.99$ V). **d** High-resolution d$I$/d$V$ spectra of TBrPP-Co on AGNR/Au(111) exhibit Kondo peaks with three different $T_K$ values ($q = 247$, 250 and 250 for 1, 2 and 3, respectively. Tip set-point: $I_0 = 1.0 \times 10^{-11}$ A, $V_0 = 0.2$ V. Spectra are vertically offset by 100 fA/mV for clarity). **e** $T_K$ as functions of adsorption site 1, 2 and 3 (Fig. 2d–f) for TBrPP-Co on AGNR/Au(111) (*green*) and on Au(111) (*blue*). Error bars are determined from the standard deviations of the Kondo fits. **f** An STM image of TBrPP-Co at the end of a molecular cluster on AGNR ($4.6 \times 3.3$ nm$^2$, $I_t = 1 \times 10^{-11}$ A, $V_t = 0.2$ V). **g** A sequence of d$I$/d$V$ spectra taken at the locations of *coloured dots* labelled 1–5 in **f** show decreasing Kondo amplitudes as the tip moves away from the molecule centre (tip set-point: $I_0 = 1.0 \times 10^{-11}$ A, $V_0 = 0.2$ V. Spectra are vertically offset by 20 fA/mV for clarity). **h** The normalized Kondo amplitude ($A/A_{max}$) as a function of distance $r$ from the molecule centre show $r^1$ dependence. **i** A zoom in STM image of a TBrPP-Co molecule ($2.0 \times 1.9$ nm$^2$, $I_t = 1 \times 10^{-11}$ A, $V_t = 0.2$ V) and corresponding d$I$/d$V$ Kondo map **j** acquired at 7 mV energy [tip set-point: $I_0 = 1.0 \times 10^{-11}$ A, $V_0 = 0.2$ V). The central part of the molecule is circled in **i**, which appears as a higher intensity in **j**

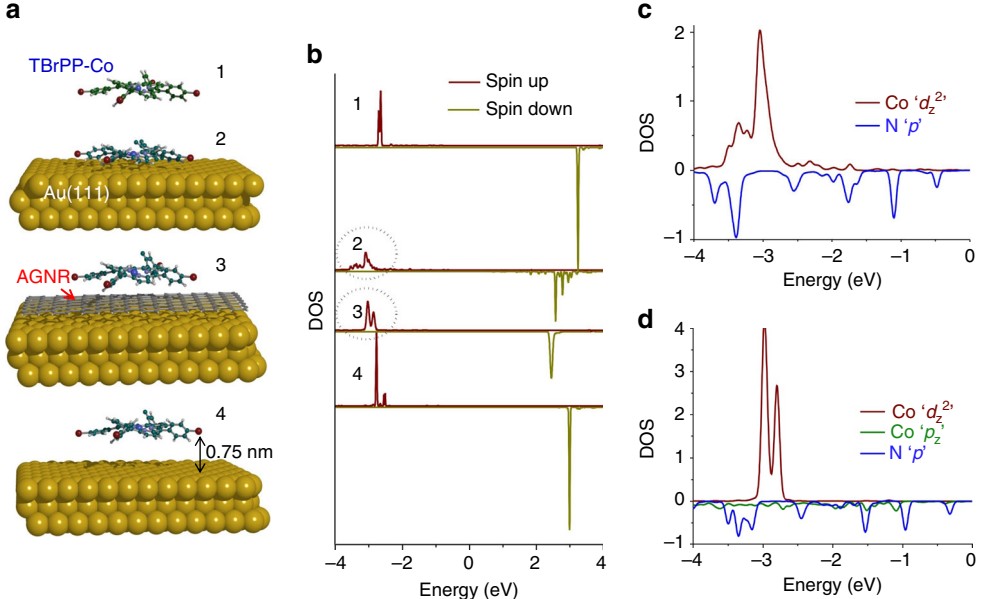

**Fig. 4** Kondo mechanism. **a** Calculated geometries of isolated TBrPP-Co (1), TBrPP-Co adsorbs on Au(111) (2), TBrPP-Co adsorbs on AGNR/Au(111) (3) and TBrPP-Co located at 7.5 Å above Au(111) surface (4) after removing the AGNR sheet in between. **b** Spin-polarized Co $d_z^2$ density of states of the TBrPP-Co corresponding to **a**: gas-phase (1), on Au(111) (2), on AGNR/Au(111) (3) and at 7.5 Å above Au(111) surface (4). **c** Atomic SP-PDOS of Co $d_z^2$ and N $p$-orbitals for TBrPP-Co adsorbs on top site of Au(111). **d** Atomic SP-PDOS of Co $d_z^2$, N $p$, and C $p_z$ orbitals in AGNR for the TBrPP-Co/AGNR/Au(111) heterostructure. Positive and negative DOS correspond to spin-up and spin-down components, respectively

d$V$ spectroscopy measured for a small bias range of ±100 mV by positioning the scanning tunneling microscope (STM) tip over the molecule's centre. These Kondo resonances are fitted with the Ujsaghy formula[21] (Supplementary Note 4), and the average Kondo temperatures ($T_K$) for the three different adsorption geometries (Fig. 2a–c) are found as 150.0 ± 5.2 K, 130.7 ± 5.1 K, and 112.4 ± 5.9 K, respectively with a small $q$ value of ~0.4 (Fig. 2h). Thus, the measured $T_K$ values exhibit a decreasing trend for the adsorption of molecules on top, bridge and hollow sites on Au(111) surface. The variation in Kondo temperature here is due to different adsorption sites, which changes the spin coupling to the host electron bath[12, 30] (Supplementary Note 5). When this molecule adsorbs on Cu(111) surface, the transfer of charge from the surface to the molecule induces a net charge in the molecule including at the hydrocarbon ligands, and as a result, the entire molecule becomes spin-active and exhibits a Kondo resonance throughout the porphyrin unit[18]. The localized nature of the observed Kondo resonance on Au(111) here (Supplementary Fig. 8) indicates that such interfacial charge transfer is not present, in agreement with the DFT and d$I$/d$V$ spectroscopy results.

**TBrPP-Co/AGNR/Au(111) heterostructures**. Next, we investigate the properties of TBrPP-Co/AGNR/Au(111) heterostructures. On AGNR, the molecule also adsorbs in a planar geometry (Fig. 3a), but it can be easily displaced even when scanning with a low tunnelling current of ~10 pA, indicating that the molecule–AGNR interaction is very weak. DFT + $U$ calculations show the TBrPP-Co preferentially positions with its Co atom located above the centre of a honeycomb on the AGNR lattice (Fig. 3b). Here, the vertical distance between the molecule and the AGNR is 4.1 Å while the AGNR is located at 3.4 Å from Au(111) surface. Thus, the TBrPP-Co is located 7.5 Å above the Au(111) surface. The d$I$/d$V$ tunnelling spectroscopy of TBrPP-Co on AGNR reveals the HOMO at −1.20 V and LUMO at +1.61 V, which gives 2.81 eV energy gap (Fig. 3c). Remarkably, unlike the molecule on Au(111), the d$I$/d$V$ intensity of TBrPP-Co on AGNR

appears almost zero and completely flat within the energy gap (Fig. 3c and Supplementary Fig. 9), indicating that the molecule is essentially electronically decoupled from the AGNR and Au(111). The slightly larger HOMO–LUMO gap of molecule on AGNR/Au(111) also points to electronic decoupling. Our DFT calculations reveal the same effect, showing sharp $d$-orbitals indicating electronic decoupling, and negligible amplitude within the gap, as discussed in a later section (Supplementary Fig. 10). Such electronic decoupling has been previously observed for C60 adsorbed on graphene[31]. Indeed, electronic decoupling between the molecule and the substrate would be expected because the molecule is adsorbed on a large bandgap AGNR, which is itself already in a physisorbed state on Au(111).

**Kondo effect in TBrPP-Co/AGNR/Au(111) heterostructures.** Surprisingly, when d$I$/d$V$ spectra are acquired at smaller bias range of ±100 mV with a higher energy resolution over the centre of the molecules on AGNRs, a clear Kondo resonance is observed around the Fermi level (Fig. 3d). Moreover, like the case of TBrPP-Co on Au(111), we find three different $T_K$ values for the individual TBrPP-Co on AGNRs (Fig. 3d). The measured $T_K$ values of 145.6 ± 7.2 K, 128.2 ± 6.1 K and 107.6 ± 3.8 K, are extremely close (~97 ± 1%) to the ones observed for the molecule directly adsorbed on Au(111) at top, bridge and hollow sites (Fig. 2h). The large $q$ values of the Kondo resonances for the molecule on AGNRs (~250) and nearly Lorentzian shapes of Kondo resonances[32] here are also in agreement with the electronic decoupling between the molecules and the substrate, as tunnelling through the Kondo resonance dominates over the continuum channel.

When a magnetic impurity is isolated by an atomically thin insulating layer, the Kondo effect can still be observed, although the Kondo resonances are substantially weaker[32, 33]. For Co atoms adsorbed on the atomically thin insulating CuN layer deposited on the metallic Cu(100) surface[34], the Kondo temperature was found to be ~2.6 K. This is just ~3% of the Kondo temperature of ~88 K when the Co atoms are directly

**Table 1 Structure parameters**

| Configurations | $E_d$ (eV) | $\Gamma$ (eV) | $U$ (eV) | Magnetic moment ($\mu_B$) | $T_K$ (K) |
|---|---|---|---|---|---|
| Isolated TBrPP-Co | −0.527 | 0 | 5.88 | −1.05 | 0 |
| TBrPP-Co/Au(111) (top site) | −0.51 | 0.363 | 5.71 | −1.045 | 150 |
| TBrPP-Co/Au(111) (bridge site) | −0.529 | 0.361 | 5.81 | −1.050 | 128 |
| TBrPP-Co/Au(111) (hollow site) | −0.54 | 0.357 | 5.72 | −1.044 | 112 |
| TBrPP-Co/AGNR/Au(111) | −0.46 | 0.330 | 5.78 | −1.049 | 142 |
| TBrPP-Co located at 7.5 Å above Au(111) | −0.562 | 0 | 5.86 | −1.048 | 0 |

Energy level ($E_d$) and width ($\Gamma$) of the Co $d_{z^2}$ orbitals, the Coulomb repulsion energy $U$ (separation between up and down levels for $d_{z^2}$), magnetic moment of the molecule and the Kondo temperature ($T_K$) computed using equation (1)

adsorbed on Cu(100)[35]. The observation of Kondo resonance of the molecules on AGNRs with almost the same $T_K$ in our measurements is surprising. To induce such a strong Kondo resonance, a large spin–exchange interaction between the magnetic moment of the molecule and free electrons from the substrate needs to take place[36]. The Kondo effect and associated temperature is a direct and sensitive probe of the spin–electron interaction strength, as it depends exponentially on the free electron density $\rho$ of the substrate and the exchange coupling to the magnetic impurity $J$, as $T_K \propto e^{-1/\rho J}$. The observed ~97% Kondo strength indicates that the spin coupling between the molecule and Au(111) surface through AGNR is nearly the same as when the molecule directly adsorbs on Au(111). This is reminiscent of the electronic transparency of graphene on Cu(111) surface, reported recently, where the copper electron density above the graphene layer was found to be dominant away from the graphene[24]. As AGNRs bandgaps make them electronically opaque, their density contribution at the Fermi level would be negligible, allowing for an even more dominant gold surface contribution above the AGNR. Our findings indicate that the AGNRs enable strong spin–electron coupling between the TBrPP-Co and Au(111) surface. Thus, they are indeed spintronically transparent, as exhibited by the strong Kondo resonances and large $q$ values.

We explore further the properties of the Kondo effect on TBrPP-Co/AGNR/Au(111) heterostructures. First, the position-dependent Kondo amplitudes are measured for a TBrPP-Co on an AGNR (Fig. 3f, g). As expected, the Kondo amplitude decreases as the tip position moves away from the centre of the molecule and exhibits a ~1 $r^{-1}$ decay[37], where $r$ is the distance from the centre of the molecule, with $T_K$ ~ 133 K (Fig. 3h). To explore the spatial distribution of the Kondo resonance within the molecule, a Kondo map (dI/dV map) is recorded at 7 meV energy (Fig. 3i, j and Supplementary Movie), which reveals that the Kondo resonance is located at the centre of the molecule where the caged Co atom and four nitrogen (N) atoms reside. The Kondo effect is observed whenever the molecule's centre is located on an AGNR independent of its width (Supplementary Fig. 11), and the measured Kondo temperatures closely follows those of the top, bridge and hollow adsorption sites on Au(111). However, if the molecule adsorbs at a bridge position between two AGNRs with its centre located directly above the Au(111) surface, but at a vertical distance of 7.5 Å, the Kondo effect is no longer observed (Supplementary Fig. 12). This further confirms that the AGNRs are the key in mediating the observed Kondo effect. The Kondo resonance also disappears when the Co centre of the molecule is located directly above H-terminated edge of the AGNR (Supplementary Fig. 13).

**Mechanism of Kondo in TBrPP-Co/AGNR/Au(111) heterostructure.** To understand the mechanism of the observed Kondo

resonance, SP-DFT calculations for the majority and minority spin (spin-up and -down) states of Co $d_{z^2}$ orbitals for four different molecular environments are performed (Fig. 4a, b). The first configuration is for an isolated TBrPP-Co where both majority and minority spin states of Co $d_{z^2}$ appear sharp and have narrow widths with high amplitudes. When the molecule adsorbs on Au(111) on the top surface site, the Co $d_{z^2}$ spin states become broadened, shift down in energy and their intensities are reduced due to hybridization with the substrate while other $d$-orbitals of the molecule remain sharp. A zoom in region of Co $d_{z^2}$ and N $p$-orbitals for occupied states for the molecule adsorbed directly on Au(111) is shown in Fig. 4c. Similar broadenings are also found for the molecule that adsorbs on the bridge and hollow sites of Au(111) with only a small change in broadening (Supplementary Fig. 14, Supplementary Note 6 and Supplementary Table 1). The broadening of the states here is directly related to the spin coupling with the electrons from Au(111). Because of its metallic nature, the electronic density of Au(111) can be found continuously near the Fermi level.

Similar broadening behaviour of Co $d_{z^2}$ orbitals is found for the TBrPP-Co adsorbed on AGNR/Au(111) substrate (Fig. 4b, d) indicating that the spin coupling is also taking place here, in agreement with the observed Kondo resonance. Next, to check the effects of AGNR on the observed spin coupling, the AGNR sheet between the molecule and the Au(111) surface is removed, and the molecule is held at 7.5 Å above the surface. Now, the spin states of the Co $d_{z^2}$ appear remarkably similar to the gas phase: both states are shifted up to the gas phase energies, and appear as narrow and sharp peaks again, as one would expect the molecule to fully decouple from the surface at that distance. This clearly highlights that without AGNR, the spin coupling between TBrPP-Co and Au(111) surface will not occur.

In order to unravel the nature of the spin–electron coupling in the observed Kondo effect, we explore the possibility of a two-orbital Kondo model[38], where the singly occupied $d_{z^2}$ level, $E_d$, and a molecular orbital $M$ with energy $E_M$, are considered. The presence of the non-magnetic molecular level $E_M$, modifies the effective density of states of the host explored by the Kondo correlations arising from $E_d$, which gives rise to corrections of the Kondo temperature. This model closely describes the electronic transport at the STM tip–molecule–surface junction although there are limitations in the interpretations. For the TBrPP-Co directly adsorbed on Au(111), the DFT + $U$ calculations reveal a change in the Co–N hybridization, which can be recognized from the occurrence of the SP projected densities of states (SP-PDOS) for Co $d_{z^2}$ and the PDOS for N $p$ states below the Fermi energy (Fig. 4c). The N PDOS peaks track the Co $d_{z^2}$ spin DOS within the TBrPP-Co (Supplementary Note 7), as one would expect from bonding geometry. However, when the molecule is on the AGNR, the N PDOS peaks are broadened (Fig. 4d) in a similar way as when on the gold surface, suggesting strong coupling to the compound host, the Au + AGNR system. Stronger coupling of

such intermediate $E_M$ state would contribute to enhancing the Kondo coupling even when the molecule is further away from the Au surface, which is consistent with our experimental observations. Similar indirect coupling mechanism via N atoms has been reported for Fe-porphyrin molecule adsorbed on Ni and Co films[39].

To further quantify this scenario, we use realistic parameters obtained from our ab initio DFT + U calculations to compute the Kondo temperatures using the Haldane formula[40],

$$k_B T_K = \frac{1}{2}\sqrt{\Gamma U}\, e^{\pi E_d(E_d + U)/\Gamma U} \tag{1}$$

where $\Gamma$ is the width of the impurity state, $E_d = E_{dz^2} - E_M$ is the Co $d_z{}^2$ effective orbital energy, and $U$ is the on-site Coulomb repulsion energy. Table 1 summarizes $\Gamma$, $E_d$ and $U$ for the four molecular environments presented in Fig. 4a as well as three molecular adsorption sites on Au(111). The calculations provide $T_K$ values of the molecules directly adsorbed on Au(111) as 150 K for the top site, 128 K for the bridge site and 112 K for the hollow sites, while on AGNR, 142 K is calculated only for the top site. Thus, these calculations qualitatively reproduce the decreasing trend of $T_K$ values for the top, bridge and hollow sites on Au (111), as well as a slightly smaller value when on AGNR + Au. Notice that although the $E_d$ values change slightly in the different environments, the change in $T_K$ is mainly produced by the changes in the orbital widths, $\Gamma$. We emphasize that although the effective orbital hybridization $\Gamma$ is modified when the molecule is on the AGNR, these changes do not produce a strong change in the DOS at the Fermi level. In that sense, the molecule remains electronically isolated and yet, the spin-flip virtual processes that result in the Kondo resonance at the Fermi level are still possible.

## Discussion

In summary, we have investigated electronic and magnetic properties of magnetic molecules adsorbed on AGNRs chemically synthesized on a Au(111) surface. We have compared the results obtained from the same type of molecules directly adsorbed on the Au(111) surface using low-temperature scanning tunnelling microscopy, and tunnelling spectroscopy experiments supplemented by SP DFT calculations on a single molecule level. The tunnelling spectroscopic measurements clearly reveal electronic decoupling of the molecules from the AGNR/Au(111) substrate, with no charge transfer; however, the detection of strong Kondo resonances reveals a robust spin coupling between the magnetic moment of the molecule and the spins of the substrate electrons mediated by the AGNR. Three different Kondo temperatures on the molecules induced by different adsorption sites on Au(111) surface are reproduced on AGNRs. Kondo resonances are known to be extremely sensitive to the spin coupling with the substrate and previous experiments show that just a slight displacement (0.6 Å) of the central Co atom of the molecule away from the surface could result in a large change (~30%) in Kondo temperature[28]. In the current experiment, however, Kondo temperatures of the molecules on AGNR are nearly identical to those directly adsorbed on Au(111) surface. This unambiguously shows that the molecules on AGNR experience almost as strong spin screening as if they were directly adsorbed on the Au(111), despite being located 7.5 Å above the metal surface. Our findings of GNRs mediating spin interactions open a new avenue of research with potential applications in spintronic, electronic and magnetic molecular sensing.

## Methods

**Sample preparation and tunnelling spectroscopy measurements**. The experiments were performed with a Createc low-temperature scanning tunnelling microscope system at ~7 K substrate temperature. Au(111) single crystal substrate was cleaned by repeated cycles of Ar + ion sputtering and annealing to 700 K. After checking the cleanliness of the Au(111) surface, DBBA molecules are deposited onto the Au(111) substrate via thermal evaporation using a custom-built Knudson cell. AGNRs were synthesized on the surface by heating the DBBA/Au(111) substrate to 400 °C under ultrahigh vacuum (UHV) environment. The sample was then transferred to the STM scanner under UHV condition and cooled down to ~7 K for the experiments. The dI/dV tunnelling spectroscopy and Kondo maps were recorded by using a lock-in amplifier, where a small voltage modulation amplitude of 10 meV with a frequency range of 700–1 kHz was added. During the dI/dV spectroscopy measurements, the STM tip remains static above the molecule, the STM feedback loop is terminated, and the spectra are recorded by sweeping the bias from the starting point to the end point (bias ramping down) and then back again (bias ramping up) to the starting point. Similarly, the dI/dV measurement along a single line was performed by terminating the STM feedback system.

**Calculation methods**. DFT calculations were carried out with the Vienna ab initio simulation package code[40] with core electrons described by the projected augmented wave method[39]. Exchange-correlation was treated in the Perdew-Burke-Ernzerhof (PBE) generalized gradient approximation (GGA)[41]. Because of the relative importance of non-bonding molecule–surface interactions, van der Waals D3 functional was used[42, 43]. The plane wave basis was expanded to a cutoff of 600 eV. The Au(111) surface was modelled by a three-layer slab with a vacuum space of 20 Å containing 500 atoms. A TBrPP-Co molecule composed of 44C, 24H, 4N, 1 Co and 4 Br atoms was placed on top of the graphene layer and the Au(111) surface. To accommodate the computation space limit, 13-unit width graphene sheet was used. A $2 \times 2 \times 1$ k point mesh was chosen for geometry optimizations with a force tolerance of 0.01 eV/Å and a Gaussian broadening of 0.02 eV was used. All of the atomic positions were relaxed except the bottom two layers of Au that were fixed to have the in-plane lattice constant of bulk Au. The geometry optimizations were converged within 2 meV per formula unit for the total energies. SP-DFT calculations for gas phase TBrPP-Co as well as the TBrPP-Co on Au(111), and TBrPP-Co/AGNR/Au(111) heterostructure were performed by using a DFT + U method[44], where a Hubbard U correction is added to account for the on-site Coulomb interactions in the localized d or f orbitals. An effective value of Coulomb interaction $U = 4.9$ eV was chosen in our calculation[18]. $\Gamma$ is the width of the $d_z{}^2$ orbital and for the multiple $d_z{}^2$ peaks, a weighted average of the peaks is calculated.

**Data availability**. The data supporting the findings reported in this article, including Supplementary Information and Supplementary Movie, are available by request from the corresponding author S.W.H.

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

## Acknowledgements

Use of the Center for Nanoscale Materials, an Office of Science user facility, was supported by the US Department of Energy, Office of Science, Office of Basic Energy Sciences, under contract no. DE-AC02-06CH11357. Y.L., K.Z.L. and S.W.H. acknowledge the support of the US Department of Energy, Office of Basic Energy Sciences grant DE-FG02-02ER46012 for STM and STS data analyses. A.D., Y.L. and A.N. acknowledge the support of the US Department of Energy Office of Basic Energy Sciences SISGR Grant DE-FG02-09ER16109 for the STM measurements and DFT calculations. K.Z.L. acknowledges the travel support of the Condensed Matter and Surface Science Program of Ohio University. S.E.U. acknowledges the support of NSF DMR grant 1508325. We gratefully acknowledge the computing resources provided by the Laboratory Computing Resource Center at Argonne National Laboratory.

## Author contributions

S.-W.H. conceived and designed the experiments; Y.L., A.D. and K.Z.L. performed the STM experiments; A.T.N. and P.Z. performed the SP-DFT calculations; A.T.N. and S.E. U. performed the Kondo calculations; B.F. provides technical support to the experiments; Y.L., K.Z.L. and S.-W.H. analysed the experimental data; and S.-W.H. and S.E.U. wrote the paper. All the authors discussed the results and commented on the manuscript.
