## [Peer Review File · Nature Communications]

Reviewers' comments:

Reviewer #1 (Remarks to the Author):

This manuscript reports the Kondo effect caused by the adsorption of TBrPP-Co molecules on Au(111) and graphene nanoribbon (GNR) grown on Au(111). The authors measure the tunneling spectra of the individual molecules in various configurations, and find that (1) the Kondo temperature (TK) depends on the adsorption site and (2) the Kondo temperature of TBrPP-Co on GNR is almost the same as that of TBrPP-Co directly bonding to Au(111). The measurements are well-designed and the results are reliable. The authors also carry out the DFT calculations to discuss the origins of these results and they claim that GNR has a novel property in the effective spin coupling responsible for the Kondo effect. Although the finding (1) is not new and has been already observed in the other molecular adsorption systems, the finding (2) is new and not yet presented as far as I know.

Two-orbital Kondo model is proposed for explaining the high TK of the molecule on GNR; the N-derived molecular orbital (MO) affects the Kondo screening and enhances TK. This mechanism is not new and has been already presented by the previous paper published in Phys. Rev. B and applied successfully to a surface Kondo system. The authors estimate TK by using the Haldane formula with the parameters obtained from the DFT calculations. The high TK is explained by the large Γ as well as the energy position of the d orbital relative to the N-derived MO. Certainly, the DOS peak of d orbital for the molecule in the gas phase is split for the molecule on GNR, but each peak is still narrow, suggesting the weak interaction and smaller Γ . Although the authors consider that GNR isolates the molecule electronically from Au(111), the large Γ is conflicted with their idea. The authors should clarify the origin of large Γ , discuss that this is just coincidence or not together with the role in the Kondo effect and translate the role of GNR into novel physics. However, the authors do not explain how to obtain the value of Γ and rationalize the origin of the large value, and their presentation on these points is quite insufficient and not convincing at all.

Thus, I am reluctant to recommend this manuscript for publication in the high-impact journal like Nature Communications.

Reviewer #2 (Remarks to the Author):

Li and coworkers report a very interesting study of the Kondo properties of Co porphyrine molecules on Au and on graphene nanoribbons on Au. Surprisingly, they find almost the same Kondo effect on both systems, even though the nanoribbon should decouple the molecule very effectively, which should completely eliminate any chance of a Kondo-type interaction with the substrate electrons.

I am completely baffled, why this effect occurs, beyond a DFT calculation, which is notoriously difficult and unreliable for such systems, no explanation is given for this effect. This is fine with me as this is an experimental paper with very clear experimental findings. Have the authors considered that the Kondo effect might not be with the substrate electrons but with movable electrons in the molecular system?

I think the paper is in fine shape and should be published with minor modifications:

Line 51 needs a reference

Line 104: replace "long-range" in "by means of a long-range dI/dV-V tunneling spectroscopy" with a different description. You are talking voltage range here and that is not clear.

Line 145: ($-97 \pm 1\%$ strength) – the word strength is weird.

Reviewer #4 (Remarks to the Author):

In this manuscript, Ngo et al. report the observation of the Kondo screening of a TBrPP-Co molecule, which has $S=1/2$, when deposited on bare Au(111) and also when separated from the Au(111) surface by a semiconducting AGNR. Using STM imaging and spectroscopy, they confirm the electronic properties of the AGNRs on the Au(111) surface and then probe both the orbital energies and low-energy Kondo screening of the molecule in a variety of configurations. DFT calculations further confirm the hybridization of the Co d orbitals with the substrate both on the bare Au(111) and on the AGNR in spite of the fact that the molecule is separated from the surface by ~ 0.7 nm.

The authors present a comprehensive set of experiments and calculations that provide convincing evidence that AGNRs can mediate spin interactions between a substrate and an adsorbate in spite of the fact that they have a large bandgap at the edges. This is an important and timely result, especially given the great technological relevance of AGNRs. Therefore, I will be happy to recommend this manuscript for publication in Nature Communications.

That being said, before I can do so there are a large number of important though relatively minor issues that the authors will first need to address. In particular, some relevant background information needs to be added to put this result in the appropriate context. Also, some statements are made early in the paper without justification or citations to relevant literature. These are properly provided later in the manuscript, but that makes going through the work confusing initially and therefore this should be corrected. While all of this will require some effort, I have no doubt that the authors will be able to do so appropriately. I list these points below:

- Most significantly, it has been demonstrated that other atomically thin insulating/semiconducting materials can mediate similar spin interactions. For example, Co atoms deposited on a Cu(100) surface capped with a single layer of copper nitride exhibit a Kondo effect [A.F. Otte et al., Nature Physics 4, 847 (2008) whose strength varies with position [J.C. Oberg et al., Nature Nanotechnology 9, 64 (2014)] even though the copper nitride has a large bandgap [C.D. Ruggiero et al., Applied Physics Letters 91, 253106 (2007)]. While in that case the Kondo temperature was substantially reduced compared to the Kondo temperature for a Co atom on bare Cu(100) [N. Knorr et al., Physical Review Letters 88, 096804 (2002)], the basic phenomenon is still the same. The authors should include this background in their discussion while rightly continuing to highlighting the fact that in their result the strength of the Kondo coupling is almost the same, thus emphasizing the important and complex role played by the AGNR.
- Since the change in the value of the LDOS is important for understanding changes in the Kondo screening, all dI/dV spectra and line maps in all relevant figures in the manuscript and Supplementary figures should be reported in absolute units (A/V, pA/mV, etc.). Furthermore, the zero on the y-axis should be clearly indicated on each panel, and any vertical offset when multiple spectra are shown in the same panel should be explicitly stated.
- For reference, the initial current (I_t) and voltage (V_t) should be reported for all topographic images and dI/dV spectra (acquired above a single point or along a line).
- On p. 2, the authors state that there is no charge transfer between the AGNR and the Au(111). They should include references for this here, as they do later in the manuscript. They may also mention that their DFT calculations are consistent with this, as they describe later as well.
- Similarly, on p. 2 the authors mention the transparency of graphene on Cu(111) but only cite the reference for this work later in the manuscript. This should be done here as well.
- On p. 3, the authors state that AGNRs are known to screen the Shockley SS; they should cite references for this.
- When discussing the screening of the SS by the AGNRs on p. 3, the authors state that “the SS intensity is reduced and its onset energy is shifted towards the surface Fermi level (i.e. 0V).” While the latter is unambiguously true, the former claim is difficult to make with STM measurements

because the overall normalization (and therefore the apparent amplitude) changes when the tip height is shifted. Did the authors measure the spectra shown in Fig. S2 with a constant tip height (i.e. by opening the feedback loop at a fixed location and then laterally displacing the tip over the different AGNRs)? If not, making a definitive claim about the intensity of the SS requires additional justification. The authors should also explicitly indicate whether there is an offset in the spectra shown in Supplementary Fig. 2a.

- On p. 3, it is clear from both experiment and theory that H termination results in an increase in the AGNR bandgap at the edges. However, the authors should provide some insight (just one or two sentences) on why the H termination causes this change.
- In Fig. 1d, although the color map makes it easy to follow the spatial evolution of the dI/dV spectra, it is difficult to interpret such plots quantitatively. The authors should show individual spectra (perhaps in the Supplementary section) for various important points (centers and edges of the different AGNRs as well as the intermediate bare Au(111) regions). These should be plotted in absolute units with the zero clearly indicated on the y-axis.
- The dI/dV spectrum of the molecule in one adsorption geometry on Au(111) is shown in Fig. 2g. The authors should comment on how much this varies among the three different binding configurations. Since the authors have performed DFT calculations for all three binding configurations, they should also compare the dI/dV spectrum with the calculated projected DOS and discuss how this varies with binding configuration.
- On p. 7, when discussing the spectrum shown in Fig. 3c, it is quite strange that there is zero intensity for dI/dV within the HOMO-LUMO gap, and yet there is a finite LDOS near the Fermi energy (as required for the Kondo resonance). One would expect the tails of the resonance to decay rapidly to zero, but as seen in Fig. 3d and Supplementary Fig. 8 this is not the case. The authors should show this more carefully, plot the tails over a broader voltage range, and show the dI/dV spectrum in absolute rather than arbitrary units.
- Also on p. 7, the authors state that a larger HOMO-LUMO gap implies electronic decoupling. However, the justification for this only becomes clear much later in the paper, when the authors discuss the DFT calculations of the d-orbital hybridization. The authors should clarify this issue at this point in the manuscript.
- On p. 7, the authors describe the Kondo temperature for the molecule on the AGNR at “equivalent adsorption sites” with respect to the Au(111). They then say that the molecule can be located above any possible surface site. Later, on p. 8 they say that the Kondo effect is observed when the molecule is anywhere on the AGNR. This is confusing and should be explained more clearly. How were the “equivalent adsorption sites” determined? How does T_K vary with position on the AGNR? Does it depend on the molecule’s location on the AGNR, the location above the Au(111), or a combination of the two?
- On p. 7, the authors refer to the ‘q’ value as being very small (0.01) on the AGNR and that this indicates that all of the tunneling goes through the Kondo resonance. In the traditional Fano formula applied to Kondo screen, q is the ratio between tunneling into the (Lorentzian) Kondo resonance vs. the continuum. Therefore, $q \sim 0$ indicates tunneling through the continuum and results in an anti-resonance, while tunneling directly through the Kondo resonance occurs as q increases and the lineshape becomes a Lorentzian when $q \rightarrow \infty$. The description here seems to suggest the opposite. The authors should clarify this.
- On p. 8, the authors discuss the reduced SS density under the AGNR. It is not obvious if the electrons participating in the Kondo screening of the molecule arise from the Au(111) 2D surface state or from the bulk states. The authors should clarify this point. If they claim that one contribution is dominant over the other, then this should be justified.
- On p. 8, the authors discuss the “spintronic transparency” of the AGNR. The strong Kondo resonance reflects the strong coupling between the molecule on the AGNR and the underlying Au(111) surface, while the q value is an indication of the pathway taken by a tunneling electron. These two need not be related: it is possible to have a large Kondo temperature and $q \sim 1$ (i.e. coherent tunneling into both

the resonance and into the metallic states). The authors should clarify this, along with the meaning of the phrase “spintronically transparent”.

- On p. 10, the authors discuss the hybridization of the Co d_{z^2} orbital. What about the other occupied Co d-orbitals? These are likely to not be strongly affected, but it would be useful to check this and comment on it.
- When discussing the Kondo effect, the authors may consider citing an excellent review: M. Ternes et al., J. Phys.: Condens. Matter 21, 053001 (2009).
- There are typos in many of the references. The authors should check these carefully.
- In Supplementary Fig. 3d, the authors show the projected DOS above H atoms in the AGNR. At what height above the surface is the PDOS calculated? For reference, a similar spectrum over the center of the AGNR should be shown. The authors should also discuss how this compares to the measured dI/dV spectra shown in Fig. 1c?
- In Supplementary Fig. 4a and b, it is difficult to see the lateral motion of the AGNR network, though once the images are overlaid it became very clear. Perhaps the authors could add a grid on both panels a and b to provide additional reference markers.
- The TBrPP-Co molecule has four-fold symmetry. However, in Supplementary Fig. 5 the appearance of two protruding lobes suggests interaction with Au(111) surface. The authors should comment on this with regard to the effect of the three different binding sites and whether a similar effect is observed on the AGNR.
- In Supplementary Fig. 6c, what does “normalized amplitude” mean? This should be clearly defined.
- In Supplementary Fig. 7, the caption for panel a should be clearly state which colors correspond to which atoms. Also, I am a little confused on how “charge density” can vary with energy? Are these plots of integrated LDOS? If so, this should be more clearly stated and the regions of integration should be clearly specified.

REVIEWERS' COMMENTS:

Reviewer #1 (Remarks to the Author):

The authors revised the paper and added more information, which makes the manuscript better than the original one. However, I am not satisfied with the revised version yet because the authors do not provide clear answer to the important point that I raised in the previous report. I previously questioned why Gamma of TBrPP-Co/GNR/Au is almost identical to that of TBrPP-Co/Au. This means that the electronic coupling is not so weak as expected. The splitting of Co d state is also indicator for the coupling. I am skeptical of the main message in abstract, "GNRs mediate effective spin-coupling while electronically isolating the molecules from the substrate, opening a way for potential applications in spintronics."

Reviewer #2 (Remarks to the Author):

The authors have answered all my questions. the paper should be published as is. Maybe the authors want to consider to invite reviewer 4 as a co-author ; -)

Reviewer #4 (Remarks to the Author):

Since all of the issues raised in the initial round of reviews have been addressed by the authors, I am happy to support publication of this manuscript in Nature Communications.

Reviewer #1 :

This manuscript reports the Kondo effect caused by the adsorption of TBrPP-Co molecules on Au(111) and graphene nanoribbon (GNR) grown on Au(111). The authors measure the tunneling spectra of the individual molecules in various configurations, and find that (1) the Kondo temperature (T_K) depends on the adsorption site and (2) the Kondo temperature of TBrPP-Co on GNR is almost the same as that of TBrPP-Co directly bonding to Au(111). The measurements are well-designed and the results are reliable. The authors also carry out the DFT calculations to discuss the origins of these results and they claim that GNR has a novel property in the effective spin coupling responsible for the Kondo effect. Although the finding (1) is not new and has been already observed in the other molecular adsorption systems, the finding (2) is new and not yet presented as far as I know.

ANS: We thank the reviewer for a succinct and accurate summary of the main points of this work. Finding (1) is an integral part of the experiment and it is necessary for proving similar Kondo resonances of molecules on AGNR and on Au(111). We appreciate also the referee's comments that our experimental measurements are "well designed and reliable" and that our findings, especially (2), are "new and not yet presented".

Two-orbital Kondo model is proposed for explaining the high T_K of the molecule on GNR; the N-derived molecular orbital (MO) affects the Kondo screening and enhances T_K . This mechanism is not new and has been already presented by the previous paper published in Phys. Rev. B and applied successfully to a surface Kondo system.

ANS: This is mostly an experimental paper and the theoretical mechanism we quote/cite here supports and explains the experimental findings. The use of a established theoretical scheme provides indeed better physical insights into the experimental observations.

The authors estimate T_K by using the Haldane formula with the parameters obtained from the DFT calculations. The high T_K is explained by the large Γ as well as the energy position of the d orbital relative to the N-derived MO. Certainly, the DOS peak of d orbital for the molecule in the gas phase is split for the molecule on GNR, but each peak is still narrow, suggesting the weak interaction and smaller Γ . Although the authors consider that GNR isolates the molecule electronically from Au(111), the large Γ is conflicted with their idea. The authors should clarify the origin of large Γ , discuss that this is just coincidence or not together with the role in the Kondo effect and translate the role of GNR into novel physics. However, the authors do not explain how to obtain the value of Γ and rationalize the origin of the large value, and their presentation on these points is quite insufficient and not convincing at all. Thus, I am reluctant to recommend this manuscript for publication in the high-impact journal like Nature Communications.

ANS: Although the effective orbital hybridization Γ is modified when the molecule is on the AGNR, these changes do not produce a strong change in the DOS at/near the Fermi level. In that sense, the molecule remains "electronically isolated" (so that the contribution to the differential conductance from these deep orbitals is negligible, for example). And yet, the spin-flip virtual processes that result in the Kondo resonance at the Fermi level are still possible, and involve the relatively sharp orbitals in the molecule. As the Kondo temperature depends *exponentially* on the Γ , a small change in that parameter would have a strong impact on T_K , while its impact on the differential conductance may still be nil. [Notice, incidentally, that Γ changes by less than 30 meV for molecule-on-Au to molecule-on-AGNR-on Au.] We have added this information in line 277 to help clarify this point.

We have also added the following sentence in line 179: *“As AGNRs bandgaps make them electronically opaque, their density contribution at the Fermi level would be negligible, allowing for an even more dominant gold surface contribution above the AGNR.”*

The Supplement (section S11) also describes further how the d_z^2 orbitals change in the different environments. The following sentence is added in line 334: *“ Γ is the width of the d_z^2 orbital and for the multiple d_z^2 peaks, a weighted average of the peaks is calculated.”*

We thank the Reviewer for raising these questions and we hope that the added text would aid the reader and provide the needed clarity in our manuscript.

Reviewer #2:

Li and coworkers report a very interesting study of the Kondo properties of Co porphyrine molecules on Au and on graphene nanoribbons on Au. Surprisingly, they find almost the same Kondo effect on both systems, even though the nanoribbon should decouple the molecule very effectively, which should completely eliminate any chance of a Kondo-type interaction with the substrate electrons. I am completely baffled, why this effect occurs, beyond a DFT calculation, which is notoriously difficult and unreliable for such systems, no explanation is given for this effect. This is fine with me as this is an experimental paper with very clear experimental findings. Have the authors considered that the Kondo effect might not be with the substrate electrons but with movable electrons in the molecular system? I think the paper is in fine shape and should be published with minor modifications:

ANS: We thank the referee for the positive comments and constructive ideas. We did consider the possibility that moveable electrons from the molecular systems would be involved, but both TBrPP-Co and AGNR do not have such electrons near the Fermi energy. Moreover, similar Kondo temperatures between the molecules on AGNR and those on Au(111) suggests the importance of the substrate electrons in the present case. We should also add that although these DFT calculations are notoriously challenging (and computing intensive), they provide striking agreement with the experimental results, as the small variations in hybridization and orbital levels are consistent with the variations in observed T_K values. More than the specific T_K values, it is clear that the trends exhibited by the experimental measurements are matched by the DFT results.

Line 51 needs a reference.

ANS: We have added the reference (ref. 24), which was originally cited as ref. 30 in line 54 (former line 51).

Line 104: replace “long-range” in “by means of a long-range dI/dV -V tunneling spectroscopy” with a different description. You are talking voltage range here and that is not clear.

ANS: The “long-range” is now changed to a “large bias-range” in line 114 (former line 104).

Line 145: ($\sim 97 \pm 1\%$ strength) – the word strength is weird.

ANS: The word “strength” has been removed.

Reviewer #4:

In this manuscript, Ngo et al. report the observation of the Kondo screening of a TBrPP-Co molecule, which has $S=1/2$, when deposited on bare Au(111) and also when separated from the Au(111) surface by a semiconducting AGNR. Using STM imaging and spectroscopy, they confirm the electronic properties of

the AGNRs on the Au(111) surface and then probe both the orbital energies and low-energy Kondo screening of the molecule in a variety of configurations. DFT calculations further confirm the hybridization of the Co d orbitals with the substrate both on the bare Au(111) and on the AGNR in spite of the fact that the molecule is separated from the surface by ~ 0.7 nm.

The authors present a comprehensive set of experiments and calculations that provide convincing evidence that AGNRs can mediate spin interactions between a substrate and an adsorbate in spite of the fact that they have a large bandgap at the edges. This is an important and timely result, especially given the great technological relevance of AGNRs. Therefore, I will be happy to recommend this manuscript for publication in Nature Communications.

ANS: We thank the referee for the positive comments and important suggestions, as well as for a careful reading of the manuscript and supplement.

That being said, before I can do so there are a large number of important though relatively minor issues that the authors will first need to address. In particular, some relevant background information needs to be added to put this result in the appropriate context. Also, some statements are made early in the paper without justification or citations to relevant literature. These are properly provided later in the manuscript, but that makes going through the work confusing initially and therefore this should be corrected. While all of this will require some effort, I have no doubt that the authors will be able to do so appropriately. I list these points below:

- Most significantly, it has been demonstrated that other atomically thin insulating/semiconducting materials can mediate similar spin interactions. For example, Co atoms deposited on a Cu(100) surface capped with a single layer of copper nitride exhibit a Kondo effect [A.F. Otte et al., Nature Physics 4, 847 (2008) whose strength varies with position [J.C. Oberg et al., Nature Nanotechnology 9, 64 (2014)] even though the copper nitride has a large bandgap [C.D. Ruggiero et al., Applied Physics Letters 91, 253106 (2007)]. While in that case the Kondo temperature was substantially reduced compared to the Kondo temperature for a Co atom on bare Cu(100) [N. Knorr et al., Physical Review Letters 88, 096804 (2002)], the basic phenomenon is still the same. The authors should include this background in their discussion while rightly continuing to highlighting the fact that in their result the strength of the Kondo coupling is almost the same, thus emphasizing the important and complex role played by the AGNR.

ANS: We are grateful to the referee for the suggestions and the important references. We have added these references (ref. 30, 33, 35, 36). An additional reference (34) is also added.

The following paragraph is added in line 166:

“When a magnetic impurity is isolated by an atomically thin insulating layer, the Kondo effect can still be observed, although the Kondo resonances are substantially weaker^{33,34}. For Co atoms adsorbed on the atomically thin insulating CuN layer deposited on the metallic Cu(100) surface³⁵, the Kondo temperature was found to be ~ 2.6 K. This is just $\sim 3\%$ of the Kondo temperature of ~ 88 K when the Co atoms are directly adsorbed on Cu(100)³⁶.”

- Since the change in the value of the LDOS is important for understanding changes in the Kondo screening, all dI/dV spectra and line maps in all relevant figures in the manuscript and Supplementary figures should be reported in absolute units (A/V, pA/mV, etc.). Furthermore, the zero on the y-axis should be clearly indicated on each panel, and any vertical offset when multiple spectra are shown in the same panel should be explicitly stated.

ANS: We have amended all relevant figures to indicate absolute units. If spectra are vertically offset, then the offset values are given and zero on ‘y’ axis are indicated in all relevant figures.

- For reference, the initial current (I_t) and voltage (V_t) should be reported for all topographic images and dI/dV spectra (acquired above a single point or along a line).

ANS: We have amended all relevant figure captions to indicate the requested information.

- On p. 2, the authors state that there is no charge transfer between the AGNR and the Au(111). They should include references for this here, as they do later in the manuscript. They may also mention that their DFT calculations are consistent with this, as they describe later as well.

ANS: A reference (ref. 23) is now added in line 49.

We have also added the following sentence, “...in agreement with our density functional theory calculations, as we describe below.” in line 50.

- Similarly, on p. 2 the authors mention the transparency of graphene on Cu(111) but only cite the reference for this work later in the manuscript. This should be done here as well.

ANS: A reference (ref. 24) concerning the transparency of graphene is now added in line 55.

- On p. 3, the authors state that AGNRs are known to screen the Shockley SS; they should cite references for this.

ANS: A reference (ref. 23) is now added in line 49.

- When discussing the screening of the SS by the AGNRs on p. 3, the authors state that “the SS intensity is reduced and its onset energy is shifted towards the surface Fermi level (i.e. 0V).” While the latter is unambiguously true, the former claim is difficult to make with STM measurements because the overall normalization (and therefore the apparent amplitude) changes when the tip height is shifted. Did the authors measure the spectra shown in Fig. S2 with a constant tip height (i.e. by opening the feedback loop at a fixed location and then laterally displacing the tip over the different AGNRs)? If not, making a definitive claim about the intensity of the SS requires additional justification. The authors should also explicitly indicate whether there is an offset in the spectra shown in Supplementary Fig. 2a.

ANS: The analysis was made from the data shown in Fig. 1C, which was measured by terminating the feedback loop along the line. Therefore it is relevant to compare the dI/dV intensities.

The following paragraph is added in method section in the main text, line 313: “During the dI/dV spectroscopy measurements, the STM tip remains static above the molecule, the STM feedback loop is terminated, and the spectra are recorded by sweeping the bias from the starting point to the end point (bias ramping down) and then back again (bias ramping up) to the starting point. Similarly, the dI/dV measurement along a single line was performed by terminating the STM feedback system.”

The dI/dV spectra in supplementary Fig. S2a are vertically offset. We have added the sentence, “spectra are vertically shifted by 0.05 pA/mV” in the caption of the supplementary Fig. S2d (former Fig. S2a).

- On p. 3, it is clear from both experiment and theory that H termination results in an increase in the AGNR bandgap at the edges. However, the authors should provide some insight (just one or two sentences) on why the H termination causes this change.

ANS: The following sentence is added in line 74; “As H binds strongly to the AGNR edges, it removes electronic density away, which effectively opens the gap locally.” Further discussion has also been included in the Supplementary section S2.

- In Fig. 1d, although the color map makes it easy to follow the spatial evolution of the dI/dV spectra, it is difficult to interpret such plots quantitatively. The authors should show individual spectra (perhaps in the Supplementary section) for various important points (centers and edges of the different AGNRs as well as the intermediate bare Au(111) regions). These should be plotted in absolute units with the zero clearly indicated on the y-axis.

ANS: We have added individual dI/dV spectra measured on bare Au(111), centers of 7, 14, and 28 AGNRs, and corresponding H terminated edges in supplementary section S2 and added new figures (Fig. S2a, b, and c). Corresponding explanations are also added.

- The dI/dV spectrum of the molecule in one adsorption geometry on Au(111) is shown in Fig. 2g. The authors should comment on how much this varies among the three different binding configurations. Since the authors have performed DFT calculations for all three binding configurations, they should also compare the dI/dV spectrum with the calculated projected DOS and discuss how this varies with binding configuration.

ANS: There are slight variations of LUMO positions; 1.64 V, 1.7 V and 1.8 V for the top, bridge, and hollow sites. Similar behaviour was also reported for the spin exchange coupling strength of Co atoms adsorbed on an CuN insulating layer by the group of Hirjibehedin [ref. 30]. We have added this information in supplementary S5 and a new figure (Fig. S5.3) is also added.

The integrated PDOS of d_z^2 also follow the trend of the observed T_K values. We have added this information in supplementary S11, and a new table (Table S11) is added.

- On p. 7, when discussing the spectrum shown in Fig. 3c, it is quite strange that there is zero intensity for dI/dV within the HOMO-LUMO gap, and yet there is a finite LDOS near the Fermi energy (as required for the Kondo resonance). One would expect the tails of the resonance to decay rapidly to zero, but as seen in Fig. 3d and Supplementary Fig. 8 this is not the case. The authors should show this more carefully, plot the tails over a broader voltage range, and show the dI/dV spectrum in absolute rather than arbitrary units.

ANS: Fig. 3c was measured at 200 G Ω tunneling resistance to avoid the mobility of the molecule on AGNR by the electric field. There is a flat density of state but only about 10 fA/mV, which is the noise level of our lock-in amplifier, and therefore it cannot be attributed to the DOS. To clarify this, we have added a simultaneously measured I-V curve corresponding to Fig. 3c for the molecule on AGNR in a new Supplementary section S14 (Fig. S14a). For comparison, the I-V curve of the molecule on Au(111) is also added.

We have also added a high resolution dI/dV spectrum (± 200 mV range) for Kondo resonance of molecule on AGNR in Fig. S14b. Here, the tunneling resistance is 20 G Ω . At this tip-height, there is a 100 fA/mV background DOS caused by tunneling into the Au substrate. All the dI/dV spectra are now presented in absolute units.

- Also on p. 7, the authors state that a larger HOMO-LUMO gap implies electronic decoupling. However, the justification for this only becomes clear much later in the paper, when the authors discuss the DFT calculations of the d-orbital hybridization. The authors should clarify this issue at this point in the manuscript.

ANS: We have slightly reworded the arguments here to make clearer connection. The following sentence is rephrased on page 7, line 148, "*Our DFT calculations reveal the same effect, showing sharp 'd' orbitals indicating electronic decoupling within the gap as discussed in a later section*".

- On p. 7, the authors describe the Kondo temperature for the molecule on the AGNR at “equivalent adsorption sites” with respect to the Au(111). They then say that the molecule can be located above any possible surface site. Later, on p. 8 they say that the Kondo effect is observed when the molecule is anywhere on the AGNR. This is confusing and should be explained more clearly. How were the “equivalent adsorption sites” determined? How does T_K vary with position on the AGNR? Does it depend on the molecule’s location on the AGNR, the location above the Au(111), or a combination of the two?

ANS: To enhance clarity, we have changed the “*equivalent adsorption sites*” to “*top, bridge, and hollow sites*” in line 161.

The confusing sentence, “*Since AGNRs can be found along different orientations on Au(111), the Co centre of TBrPP-Co adsorbed on AGNRs can be located above corresponding surface sites of Au(111).*” is removed.

In line 193, we replace the sentence “*the Kondo effect is observed when the molecule is anywhere on the AGNR*” with the following sentences, “*The Kondo effect is observed whenever the molecule’s centre is located on an AGNR independent of its width*” and “*the measured Kondo temperatures closely follow those of the top, bridge, and hollow adsorption sites on Au(111)*”.

- On p. 7, the authors refer to the ‘q’ value as being very small (0.01) on the AGNR and that this indicates that all of the tunneling goes through the Kondo resonance. In the traditional Fano formula applied to Kondo screen, q is the ratio between tunneling into the (Lorentzian) Kondo resonance vs. the continuum. Therefore, $q \sim 0$ indicates tunneling through the continuum and results in an anti-resonance, while tunneling directly through the Kondo resonance occurs as q increases and the lineshape becomes a Lorentzian when $q \rightarrow \infty$. The description here seems to suggest the opposite. The authors should clarify this.

ANS: We are really grateful to the referee for pointing this out. We have mistakenly quoted the values of ‘q’. We apologize for this terrible oversight. We now report the correct ‘q’ value in line 160, which agree with our interpretation and the usual convention for the Ujsaghy fits.

- On p. 8, the authors discuss the reduced SS density under the AGNR. It is not obvious if the electrons participating in the Kondo screening of the molecule arise from the Au(111) 2D surface state or from the bulk states. The authors should clarify this point. If they claim that one contribution is dominant over the other, then this should be justified.

ANS: It is not our intention to probe whether the surface or bulk states dominate the Kondo screening in gold. It would appear they both participate, as the contribution from the SS is likely suppressed in the presence of the AGNR, and yet the Kondo screening is strong. We have removed the sentence “*By considering a reduced SS density underneath AGNRs caused by screening*” and reworded this section to provide a clear discussion.

- On p. 8, the authors discuss the “spintronic transparency” of the AGNR. The strong Kondo resonance reflects the strong coupling between the molecule on the AGNR and the underlying Au(111) surface, while the q value is an indication of the pathway taken by a tunneling electron. These two need not be related: it is possible to have a large Kondo temperature and $q \sim 1$ (i.e. coherent tunneling into both the resonance and into the metallic states). The authors should clarify this, along with the meaning of the phrase “spintronically transparent”.

ANS: We have attempted to clarify this section. We are of course trying to provide an intuitive discussion on the confluence of strong Kondo signals. We hope the current phrasing provides a better picture that is consistent with the DFT calculations and the observations.

The following sentence is modified in line 179: *“As AGNRs bandgaps make them electronically opaque, their density contribution at the Fermi level would be negligible, allowing for an even more dominant gold surface contribution above the AGNR. Our findings indicate that the AGNRs enable strong spin-electron coupling between the TBrPP-Co and Au(111) surface. Thus they are indeed spintronically transparent, as exhibited by the strong Kondo resonances and large ‘q’ values.”*

- On p. 10, the authors discuss the hybridization of the Co d_{z^2} orbital. What about the other occupied Co d-orbitals? These are likely to not be strongly affected, but it would be useful to check this and comment on it.

ANS: We have added the following sentence in line 232: *“the Co ‘ d_z^2 ’ spin states become broadened, shift down in energy, and their intensities are reduced due to hybridization with the substrate while other ‘d’ orbitals of the molecule remain sharp”.*

- When discussing the Kondo effect, the authors may consider citing an excellent review: M. Ternes et al., J. Phys.: Condens. Matter 21, 053001 (2009).

ANS: We thank the referee for this reference. It is now added as the ref. 22 in line 44.

- There are typos in many of the references. The authors should check these carefully.

ANS: We have corrected all typos.

- In Supplementary Fig. 3d, the authors show the projected DOS above H atoms in the AGNR. At what height above the surface is the PDOS calculated? For reference, a similar spectrum over the center of the AGNR should be shown. The authors should also discuss how this compares to the measured dI/dV spectra shown in Fig. 1c?

ANS: The DOS is projected from the H atom at a height of 3.3Å to the Au surface underneath. This is now stated in the supplementary S3. We have also included a figure that compares the PDOS of the H atoms with that of an interior C atom in the AGNR in Fig. S3. A discussion concerning DOS of H-edge and C atoms at the centre of the AGNR is also added in Supplementary S3.

- In Supplementary Fig. 4a and b, it is difficult to see the lateral motion of the AGNR network, though once the images are overlaid it became very clear. Perhaps the authors could add a grid on both panels a and b to provide additional reference markers.

ANS: We have added grids to the Supplementary Fig. S4a and S4b.

- The TBrPP-Co molecule has four-fold symmetry. However, in Supplementary Fig. 5 the appearance of two protruding lobes suggests interaction with Au(111) surface. The authors should comment on this with regard to the effect of the three different binding sites and whether a similar effect is observed on the AGNR.

ANS: The appearances of the molecules here are dominated by the symmetry of the molecular orbital. The HOMO of the molecule has a two-lobe shape, and it is energetically located at ~ -1 V. The LUMO has a symmetric four-lobe shape, and it is energetically located at $\sim +1.7$ V. We have added the dI/dV maps of HOMO and LUMO on Au(111) measured at -1V and +1.7 V energy together with corresponding STM images in Fig. S5.1 a, b, c, and d in supplementary S5.

The observed two-lobe and 4-lobe shapes of the molecule are contributed by the HOMO and LUMO orbitals if the bias used for tunneling exceed the respective energies. This does not change with the adsorption site on Au(111) surface nor on AGNR, because it is the orbital shape of the molecule. We

have added 2-lobe and 4-lobe structures of the molecules for the top, bridge and hollow adsorption sites on Au(111) as well as on AGNRs in Fig. S5.2.

- In Supplementary Fig. 6c, what does “normalized amplitude” mean? This should be clearly defined.

ANS: The Kondo amplitudes are normalized by dividing with the maximum amplitude measured at the center of the Co atom of the porphyrin unit. (This sentence is added in Supplementary S6.)

- In Supplementary Fig. 7, the caption for panel it should be clearly state which colors correspond to which atoms. Also, I am a little confused on how “charge density” can vary with energy? Are these plots of integrated LDOS? If so, this should be more clearly stated and the regions of integration should be clearly specified.

ANS: Atom labels are added in Fig. S7. These plots are integrated LDOS, as it is now mentioned in the Fig. S7 caption. The regions of integrations are also described.

Answers to the Referees

Reviewer #1

The authors revised the paper and added more information, which makes the manuscript better than the original one. However, I am not satisfied with the revised version yet because the authors do not provide clear answer to the important point that I raised in the previous report. I previously questioned why Gamma of TBrPP-Co/GNR/Au is almost identical to that of TBrPP-Co/Au. This means that the electronic coupling is not so weak as expected. The splitting of Co d state is also indicator for the coupling. I am skeptical of the main message in abstract, "GNRs mediate effective spin-coupling while electronically isolating the molecules from the substrate, opening a way for potential applications in spintronics."

ANS: We have removed the sentence, "*while electronically isolating the molecules from the substrate*" from the abstract. We have also added the following sentence in line 279, "*...there are limitations in the interpretations*". As referee 2 pointed out in the first referee round, this is an experimental manuscript and theory is just to support experimental finding.

Reviewer #2

The authors have answered all my questions. the paper should be published as is. Maybe the authors want to consider to invite reviewer 4 as a co-author ;-)

Reviewer #4

Since all of the issues raised in the initial round of reviews have been addressed by the authors, I am happy to support publication of this manuscript in Nature Communications.